# EndoVAscular treatment and ThRombolysis for Ischemic Stroke Patients (EVA-TRISP) registry: basis and methodology of a pan-European prospective ischaemic stroke revascularisation treatment registry

Annika Nordanstig [1,2] Sami Curtze,[3] Henrik Gensicke,[4] Sanne M Zinkstok,[5] Hebun Erdur,[6] Camilla Karlsson,[1] Jan-Erik Karlsson,[1,2] Nicolas Martinez-Majander,[3] Gerli Sibolt,[3] Philippe Lyrer,[4] Christopher Traenka,[4] Merih I Baharoglu,[5] Jan F Scheitz,[7] Nicolas Bricout,[8] Hilde Hénon,[8] Didier Leys,[8] Ashraf Eskandari,[9] Patrik Michel,[9] Christian Hametner,[10] Peter Arthur Ringleb,[10] Marcel Arnold,[11] Urs Fischer,[11] Hakan Sarikaya,[11] David J Seiffge [11] Alessandro Pezzini,[12] Andrea Zini,[13] Visnja Padjen,[14] Dejana R Jovanovic,[14,15] Andreas Luft,[16] Susanne Wegener,[16] Lars Kellert,[17] Katharina Feil [18] Georg Kägi,[19] Alexandros Rentzos,[20,21] Kimmo Lappalainen,[22] Ronen R Leker,[23] Jose E Cohen,[24] John Gomori,[25] Alex Brehm,[26] Jan Liman,[26] Marios Psychogios,[27,28] Andreas Kastrup,[29] Panagiotis Papanagiotou,[30,31] Jan Gralla,[32] Mauro Magoni,[33] Charles B L M Majoie,[34] Georg Bohner,[35] Ivan Vukasinovic,[36] Vladimir Cvetic,[37,38] Johannes Weber,[39] Zsolt Kulcsar,[40] Martin Bendszus,[41] Markus Möhlenbruch,[41] George Ntaios,[42] Eftychia Kapsalaki,[43] Katarina Jood,[2,44] Christian H Nolte [6] Paul J J Nederkoorn,[5] Stefan Engelter,[45,46] Daniel Strbian,[3] Turgut Tatlisumak,[47,48] on behalf of EVA-TRISP Investigators

AN, SC, HG, SMZ, HE and CK are joint first authors.

For numbered affiliations see end of article.

**Correspondence to**
Dr Annika Nordanstig;
annika.nordanstig@vgregion.se

## ABSTRACT

**Purpose** The Thrombolysis in Ischemic Stroke Patients (TRISP) collaboration was a concerted effort initiated in 2010 with the purpose to address relevant research questions about the effectiveness and safety of intravenous thrombolysis (IVT). The collaboration also aims to prospectively collect data on patients undergoing endovascular treatment (EVT) and hence the name of the collaboration was changed from TRISP to EVA-TRISP. The methodology of the former TRISP registry for patients treated with IVT has already been published. This paper focuses on describing the EVT part of the registry.

**Participants** All centres committed to collecting predefined variables on consecutive patients prospectively. We aim for accuracy and completeness of the data and to adapt local databases to investigate novel research questions. Herein, we introduce the methodology of a recently constructed academic investigator-initiated open collaboration EVT registry built as an extension of an existing IVT registry in patients with acute ischaemic stroke (AIS).

**Findings to date** Currently, the EVA-TRISP network includes 20 stroke centres with considerable expertise in EVT and maintenance of high-quality hospital-based registries. Following several successful randomised

## Strengths and limitations of this study

► The EndoVAscular treatment and ThRombolysis for Ischemic Stroke Patients (EVA-TRISP) collaboration offers a platform to pool individual patient data from prospective registries of patients with ischaemic stroke undergoing revascularisation therapies.

► The large sample size (currently >13000 endovascular treatments (EVTs) from 20 centres), high level of completeness of data and standardised data ascertainment are strengths of EVA-TRISP.

► EVA-TRISP will provide data from everyday clinical practice and address clinically important questions about safety and outcomes of patients with ischaemic stroke treated with EVT who are not covered by randomised controlled trials.

► Data are derived from registries that are neither monitored nor randomised. There will be no control group without EVT, which disallows the assessment of effectiveness of EVT in study populations.

controlled trials (RCTs), many important clinical questions remain unanswered in the (EVT) field and some of them will unlikely be investigated in future RCTs. Prospective

registries with high-quality data on EVT-treated patients may help answering some of these unanswered issues, especially on safety and efficacy of EVT in specific patient subgroups.

**Future plans** This collaborative effort aims at addressing clinically important questions on safety and efficacy of EVT in conditions not covered by RCTs. The TRISP registry generated substantial novel data supporting stroke physicians in their daily decision making considering IVT candidate patients. While providing observational data on EVT in daily clinical practice, our future findings may likewise be hypothesis generating for future research as well as for quality improvement (on EVT). The collaboration welcomes participation of further centres willing to fulfill the commitment and the outlined requirements.

## INTRODUCTION

Timely recanalisation improves outcomes in patients with acute ischaemic stroke (AIS).[1 2] Safety and efficacy of recanalisation strategies, namely intravenous thrombolysis (IVT) and more recently endovascular treatment (EVT) (including mechanical thrombectomy with various techniques and devices in AIS patients with anterior circulation large artery occlusions), have been well documented in several randomised controlled trials (RCTs).[3–9] A meta-analysis of five RCTs revealed an average 2.5-fold reduction in disability through EVT in large vessel occlusions compared with standard care, including IVT.[10] Early recanalisation is currently the cornerstone of acute stroke treatment with increasing use globally. This benefit is substantially higher with earlier achievement of recanalisation and diminishes with longer onset-to-treatment intervals.[2 11 12] Previous research has explicitly shown that IVT with alteplase within 4.5 hours of symptom onset improved AIS patient outcomes.[13]

Following the results of these RCTs, EVT is recommended as standard of care in patients with intracranial large vessel occlusion in several guidelines.[14–16] Consequently, health systems all over the world have adapted and identify and quickly transfer eligible patients to centres offering EVT. Simultaneously, capacity, logistics, know-how and 24/7 coverage were developed to cope with the quickly increasing demand for this intervention. Moreover, two recent RCTs showed benefit with EVT in patients treated up to 16 or 24 hours after stroke onset, given that presence of a considerable amount of salvageable brain tissue had been demonstrated with appropriate imaging methods.[17 18]

Relevant questions in the daily clinical work of treating stroke patients suffering from large vessel occlusions remain. First, EVT-RCTs included a highly selective patient population. This increased the chances to demonstrate efficacy and to exclude patients who presumably had a low chance for a favourable outcome and patients who had a high risk for serious complications. Second, the seven published EVT trials[3 5–9 19] analysed altogether only included 1754 randomised patients (of whom 869 underwent EVT) with the single smallest trial including only 65 patients (of whom 33 underwent EVT).[8] Usually, after a novel treatment is proved effective, a new wave of assumptions and extrapolations for treating a broader domain of

patients begins. As all these excluded patient subgroups cannot be studied in future RCTs, in most cases, judgements for treating or not treating with EVT will be based on limited knowledge. Some remaining questions will eventually be answered with a long delay, but some will never be answered in forthcoming RCTs. However, stroke physicians keep facing patients where evidence-based data do not explicitly contribute to decision making for these individuals, in which available treatments may very well have a potential benefit as well. Here, prospective high-quality multicentre registries including large numbers of patients representing many subgroups, not included or not separately analysed within RCT settings, may offer helpful information for basing clinical judgements while being aware that the level of certainty will not reach that gained from RCTs. These registries may also give strong clues on how trial results are implemented to clinical practice and how daily practice safety and efficacy levels match with those gained in RCTs. Furthermore, registry-based data deliver hints in generating new and adequate hypotheses for future RCTs. Another important aspect is the recently developing new field of clot property research: interested centres can collect detached clots and ship them to laboratories where macroscopic and microscopic properties of the clot coupled with clinical data can be further investigated and may open new avenues in understanding stroke mechanisms. Lastly, quality is a central indicator in healthcare, and registry-based data can be used in comparisons and for improving individual centre acute stroke care pathways. As a prerequisite, such data have to be based on well-maintained registries containing a large number of detailed, clearly defined and well-characterised variables. The Endovascular Treatment and Thrombolysis for Ischemic Stroke Patients (EVA-TRISP) registry aims at meeting these prerequisites. Using our decade-long experience from the multinational TRISP registry,[20] we are now aiming to build a prospective multinational registry of AIS patients treated with EVT including detailed clinical, laboratory and imaging data for future analyses. We are presenting herein the current versions of the clinical and imaging database items of the EVA-TRISP registry. Additionally, we will discuss a selection of specific related topics.

## AIMS OF THE EVA-TRISP REGISTRY

The major aim of EVA-TRISP is to address clinically important questions about safety and outcomes in AIS patients treated with IVT and/or EVT that are not covered by RCTs. The idea of EVA-TRISP is that experienced stroke centres with a record and expertise in both: (1) usage of IVT and/or EVT and (2) maintenance of hospital-based stroke databases pool their data. In addition to the characteristics of the EVA-TRISP centres stated previously, an advantage of EVA-TRISP is the availability of more additional variables than in other large-scale registries and the commitment by the collaborators to: (1) submit accurate and complete data and to (2) be

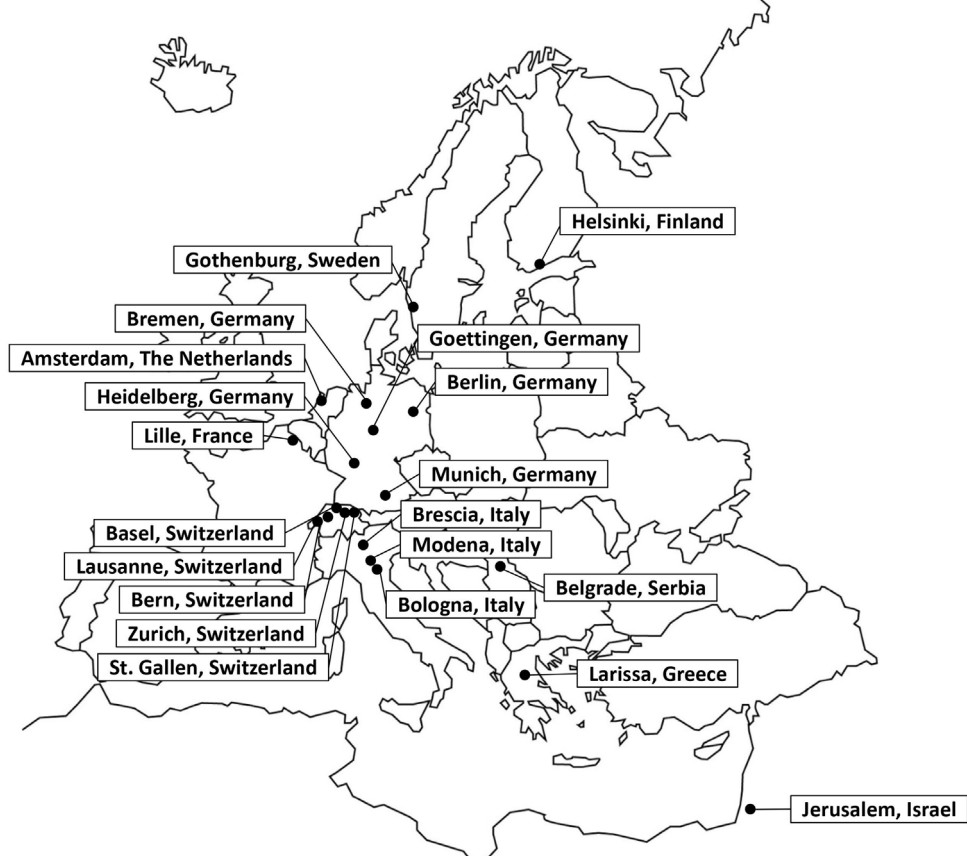

**Figure 1** EVA-TRISP centres.

willing to adapt the local databases and quickly add new variables retrospectively and prospectively.

## COHORT DESCRIPTION

The TRISP collaboration was a concerted effort initiated in 2010 by 11 European stroke centres with the purpose to address clinically relevant research questions about the effectiveness and safety of IVT, and currently, 20 stroke centres from nine different countries participate in the collaboration (see figure 1 and online supplemental appendix 1 for a list of member sites and investigators). As the collaboration also aims to prospectively collect granular and high-quality data on all consecutive stroke patients undergoing EVT, the name is changed from TRISP to EVA-TRISP.

EVA-TRISP (former TRISP) operates as an independent, non-profit, investigator-driven open platform that focuses on generating high-quality data for clinical research purposes. The EVA-TRISP research initiatives are characterised by informal, project-driven and relaxed work processes based on a high level of mutual trust and understanding between participating collaborators, and no formal scientific leadership committees have so far been implemented or deemed necessary. Internal communication principally takes place via email and teleconferences. The EVA-TRISP collaboration also hosts an annual face-to-face meeting during the European

Stroke Organisation (ESO) conference, and this forum is used for overall strategic planning and major decisions. The collaboration welcomes new collaborators and project proposals from all stroke centres that fulfil the requirements as stated further. The collaboration particularly aims at supporting young researchers. Thus, with the exception of the very first paper,[21] first authors of the publications generated by the TRISP invariably have been young stroke physicians or PhD students. The methodology of the TRISP registry has previously been published,[20] and currently, data on more than 18 000 IVT-treated patients are available in the registry. This paper focuses on describing the EVT part of the registry.

The EVA-TRISP registry aims to prospectively collect granular and high-quality data on all consecutive AIS patients undergoing both IVT and/or EVT. The overall purpose is to provide a means to address clinically important research questions on the safety and effectiveness of IVT and/or EVT in AIS patients that are typically not covered by RCTs or single-centre research initiatives. Furthermore, the EVA-TRISP aims at providing a large data source for various stroke care quality improvement initiatives. All EVA-TRISP centres have a proven track record for delivering high-quality and high-volume stroke patient care. Every EVA-TRISP centre offers stroke management that fulfils the criteria of stroke centres or stroke units as proposed by the ESO.[22] The simple idea

> **Box 1  Universal standards and requirements for the databases and centres contributing to the EVA-TRISP registry\***
>
> ► Prospective registry of consecutive patients with systematic check-up of missing cases.
> ► Comprehensive collection of baseline characteristics according to consensus definitions stated in this and the previous methodology papers.
>   Prospective assessment of haemorrhagic complications (symptomatic intracerebral haemorrhage according to the European Cooperative Acute Stroke Study (ECASS) ECASS II criteria) and functional outcome at 3 and 12 months (according to the modified Rankin Scale; either telephone interview, postal questionnaire or follow-up visit).
> ► Approval of institutional review board to maintain the respective endovascular treatment (EVT) database and to obtain 3-month and 12-month follow-up data.
> ► EVA-TRISP centres are comprehensive stroke centres with high-volume EVT applications – typically university hospitals or closely affiliated to university hospitals.
> ► Treatment of acute ischaemic stroke patients with EVT according to guidelines valid at the relevant time or documentation of deviation therefrom.
>
> \*EVA-TRISP welcomes participation and project proposals of further centres fulfilling the commitment and the outlined requirements.

of the EVT part of the EVA-TRISP registry is that experienced stroke centres with expertise in both EVT implementation and in the maintenance of hospital-based EVT databases pool their data together. An advantage of the EVA-TRISP registry is the availability of substantially more variables than in other large-scale registries. We strive for a commitment by the collaborators to provide data of high accuracy and completeness as well as towards a willingness among collaborators to swiftly adapt the local databases by adding new variables of interest. This enables a potential for explorative insights in the putative prognostic importance of variables with unknown influence on outcome or risk of complications, such as symptomatic intracranial haemorrhage (sICH). Strengths and limitations of the EVA-TRISP registry in general and compared with other existing EVT registries are discussed further (please see Discussion).

The participating centres have all agreed to fulfil the prerequisites that are summarised in box 1. Participation in other registries does not preclude participation in the EVA-TRISP registry. A standard database template has been developed by an international expert group consisting of stroke physicians and scientists in collaboration with all members, leading to the current standard version of the registry (see online supplemental appendix 2 – registry data elements), which has been agreed on by all EVA-TRISP member sites. This comprehensive dataset includes over 110 items and covers a wide range of data elements, including demographics, prestroke health information, acute phase management and long-term outcomes including 3 months and 1 year modified

Rankin Scale (mRS) as well as detailed laboratory data and imaging findings. An add-on imaging repository is currently under preparation and will be implemented within the near future. Electronic medical records used at all member sites allow for additional variables to be collected quickly and reliably when deemed necessary for new projects.

### Target population

All AIS patients designated for EVT and in whom interventionalists gained arterial access are within the target population. Therefore, this registry also may include patients with misdiagnosis, already recanalised leading to premature interruption of the procedure, unsuccessful attempts to recanalise and other unforeseeable conditions. Inclusion of patients is not limited to certain EVT techniques. The registry will include also patients who receive intra-arterial thrombolysis even without mechanical thrombectomy.

### Data collection and definitions

Data on the characteristics of patients are collected prospectively by all participating centres using standardised definitions and a standardised form. Not all centres have to provide data on all variables but have given a commitment to add missing variables retrospectively, if considered relevant to answer a specific research question.

The dataset includes patient demographics, history, prehospital information, admission data, details on acute interventions, stroke unit and/or intensive care information, discharge, rehabilitation, outcome data (3 months and 1 year outcomes measured by mRS), as well as detailed laboratory test results, vital signs and imaging findings (see online supplemental appendix 2). Risk factors and stroke aetiology will be determined according to standard approaches across centres.[20]

Moreover, neuroimaging findings before and after treatment are systematically ascertained and comprise imaging modality (CT vs MRI) as well as specific imaging findings such as hyperdense artery sign, presence and extent of early ischaemic signs, site of vessel occlusion, collateral status, presence of tandem occlusion of ipsilateral carotid artery, recanalisation status immediately after EVT and on follow-up imaging (quantified according to modified treatment in cerebral ischaemia score),[23] white matter disease severity, presence and burden of cerebral microbleeds.

All patients are monitored for occurrence of haemorrhagic transformation. Follow-up imaging usually takes place close to 24 hours after treatment or earlier in case of clinical worsening. Some centres perform follow-up imaging only in cases with clinical worsening. The definition of sICH is in accordance with the definition used in the European Cooperative Acute Stroke Study II ('an intracranial hemorrhage was defined as symptomatic if the patient had clinical deterioration causing an increase in the National Health Stroke Scale

(NIHSS) score of more than or equal to four points and if the hemorrhage was likely to be the cause of the clinical deterioration').[24] The majority of centres additionally evaluate type of haemorrhagic transformation (haemorrhagic infarction and parenchymal haemorrhage), indicate whether the bleeding occurred remotely from the infarcted area and document sICH according to definitions used in the National Institute of Neurological Disorders and Stroke Trial I+II, SITS-MOST and ECASS III definitions.[25–27] Functional outcomes at 3 months and 1 year are assessed using the mRS. The mRS is obtained by telephone calls, postal/electronical questionnaires or outpatient visits. If patients cannot be interviewed, close relatives, nurses or family doctors are asked for disability status.

We are currently exploring technical, legal and ethical as well as financial backgrounds for establishing an electronic registry to one of the member centres that would allow direct data insertion from each centre and holding the registry compactly in one single file. Otherwise, each centre will maintain their own registry within their own electronic system, and data will always be transferred without personal identifiers for each single analysis looking at one single aspect and leading to one mutual international publication. Establishing and maintaining such a large database with complete compliance to rules and safety measures is a costly procedure and requires long-term funding. This option is now under exploration.

Future plans in the registry include the addition of a neuroradiology imaging bank that will enable detailed image analysis of patients treated with IVT and/or EVT. The imaging bank will provide 'real-world' diagnostic neuroradiology and, used in combination with detailed clinical information from the EVA-TRISP registry, analysis of these imaging data will help to: (1) create standardised imaging protocols in acute stroke (ie, defining optimal threshold for perfusion parameters), (2) identify new (ie, a collateral score for the posterior circulation) and validate published (ie, different collateral scores for the anterior circulation) imaging outcome predictors and imaging-based selection tools for reperfusion therapies, (3) assess the generalisability of RCT results to subgroups of patients who would have been excluded based on imaging criteria (ie, baseline Alberta Stroke Program Early CT score (ASPECTS) under five, extracranial vessel pathologies), (4) improve automated analysing techniques (ie, machine and deep learning algorithms), and (5) enhance the accuracy of outcome prediction of different clinical and imaging parameters by implementing new imaging outcomes (e.g. infarct volume, recanalisation status). Neuroimages from TRISP/EVA-TRISP patients since 2015 will be pooled centrally. All imaging modalities (non-contrast CT, CT angiography, CT perfusion, MRI, MRA, MR perfusion and digital subtraction angiography) at baseline and follow-up (up to 3 months after stroke onset) will be eligible for analysis. Image analyses will be performed blinded to clinical information and treatment decisions and undergo a central systematic

| Table 1 | EVA-TRISP centres, time period, number of endovascular treatments done and population-base for EVT (in alphabetical order) |
| --- | --- |
| City | No of stroke EVT (January 2015 to December 2019) |
| Amsterdam | 864 |
| Basel | 413 |
| Belgrade | 136* |
| Berlin | 480† |
| Bern | 1422 |
| Bremen | Estimation: 200/year |
| Brescia | 412 |
| Bologna | 395 |
| Goettingen | 396 |
| Gothenburg | 1097 |
| Heidelberg | 1500 |
| Helsinki | 796† |
| Jerusalem | 249 |
| Larissa | – |
| Lausanne | 732 |
| Lille | 1806 |
| Modena | 489 |
| Munich | 600 |
| St. Gallen | 490 |
| Zurich | Estimation: 500 |

*January 2018–December 2019.
†November 2015–December 2019.

re-evaluation using a specified case report form including all predefined imaging variables.

Stroke-specific image analysing software (ie, OSIRIX medical imaging viewer, Quantomo for semiautomated volumetric analysis and Rapid Processing of Perfusion and Diffusion for perfusion analysis) could be used. A detailed case report form for image analysis will be designed in review with all collaborators.

Lastly, EVA-TRISP investigators prepared a detailed standard operating procedure (SOP) to ascertain that all members collecting data are well aware of standard interpretations and follow identical steps to avoid unnecessary heterogeneities or individual-borne differences. The numbers of recruited patients in each participating centre during the study period until the end of year 2019 are reported in table 1 along with the population each centre is covering for EVT.

### Data elements and completeness

A detailed database is aimed to facilitate the investigation of various current and future topics. Data elements are listed in detail in online supplemental appendix 2. Over time, new data elements may become necessary for new individual projects. Should this occur, investigators will

quickly supplement the missing variables. Patient age, sex, admission NIHSS score, recanalisation status before and after thrombectomy and 3 months outcome measured by mRS are obligatory data items and must be present for all patients (otherwise a patient is not eligible to be included to the registry). In general, missing data for any variable or patient must not exceed 10%.

## Data sources

All paper-based or electronic patient files including laboratory values and imaging data will be used to capture the EVA-TRISP registry data points. Based on these different sources alongside repeated clinical evaluations undertaken by the dedicated EVA-TRISP collaborator, registry data points that initially remain missing during the early stroke treatment process will in the majority of cases be possible to reconstruct and thereafter reported to the local registry (eg, NIHSS scores). In most cases, the local EVA-TRISP investigators form the local stroke team and will be actively seeing the patients already in the emergency room and/or at their own stroke units and can therefore guarantee completeness of data in most cases.

## Quality control

Quality control is another crucial step in multicentre large-sized registries since missing data are a frequent problem impairing the reliability and generalisability of registry-borne data. The EVA-TRISP registry is different in this sense because data are not yet directly collected to a central registry, but each centre collects their own data to their own institutional registry according to a standard harmonised database item list and SOP. Thus, clear variable definitions are easily available for the user when data are entered. It is the responsibility of the local EVA-TRISP collaborator (usually a senior stroke physician) to check and account for the validity and completeness of all data points introduced to the local registry. Therefore, missing data are expected to be very low. Furthermore, all centres have agreed to include all consecutive patients attempted with an EVT, and all centres will undertake frequent spot checks against internal hospital administrative systems covering EVT procedures as to not leave any patient out of the registry. Therefore, our registry data will likely include all EVTs performed within a region and population, practically equalling to a population-based study, although being hospital based, because EVT is usually available only at stroke centres serving a predefined region and the inhabitant population. Most of the required data come from routine procedures that are standardly collected and recorded in stroke patient care pathways as part of the clinical routine, and therefore, these data points are almost always retrievable. In the subsequent quality control process, the data files from the individual participating centres are merged into a single file for further maintenance analyses. Pseudonymised individual centre data are sent to the centre leading the specific project using encrypted transfer protocols. The subsequent data management of the merged database

will implement checks for missing data along with checks for range, consistency and illogical data (eg, NIHSS score cannot be minus or over 42 points and can only be full points and not decimals; patient age at stroke onset can be only in digits and is expected to be from 16 years and very rarely over 100 years). Also other procedures regarding quality check will be implemented at milestone points, such as comparing the performance of data reporting among centres.

## Registry size and duration

The registry will include all EVT patients from all member sites. We anticipate that the absolute numbers and proportions of EVT-treated AIS patients will be increasing over time and annual inserts will exceed thousands shortly. The registry size is not limited. One anticipated strength of this registry is the high patient number together with detailed information on each patient that will allow us to look at many issues that are not feasible to investigate within RCTs or even merged data from all RCTs because of the fact that they include fairly small patient numbers. The use of the registry will be launched after 5000 patients' data have been inserted and adequately quality checked. Similarly, this registry will be used as long as EVT is a viable option in stroke treatment. The unlimited time span requires careful evaluation by ethics committees. If the consortium decides to end the registry, each centre's data will be adequately returned to the owners, and the registry data will be deleted achieving an absolutely non-retrievable condition according to technical SOPs of the registry-holding centre. Thereafter, each individual centre will be free to decide how to proceed with their own datasets. Similarly, if a centre decides to resign from the registry, their data will be adequately returned and will be deleted from the main database file after confirmation that the data are safely received by the local principal investigator. While all patient-related data including clinical, laboratory and imaging data are completely anonymised, each centre will keep a key file within their local electronic hospital system with patient identifiers matching to the patient code in the registry (eg, if necessary to go back to patient files). This approach is compliant with current principles and is the SOP worldwide.

## Lead of a single project and authorship principles

The researcher or researchers—usually one or two (rarely three)—who originally present the idea and render the analysis proposal make(s) the initiation by drafting a standard 1–2 pages draft stating a clear hypothesis and statistical plan summarising the project (project proposal). The proposal is circulated to all member centres and discussed for scientific content and feasibility enriched with input from a large expert community. After that, the enriched plan along with the list of data items required is recirculated. If a centre agrees to participate, the centre contributes data within the in-advance agreed time frame on all consecutive patients with the key variables

of interest. For some projects, retrospective collection of data is required. The original proposal makers are entitled to the first and senior authorships. Coauthorships are distributed according to contributions. This includes mere quantitative means (ie, number of patients contributed) and quality of data (eg, completeness; considered high across EVA-TRISP centres), handling and pooling of the multicentre data; maintenance of the pooled data set (including data cleaning), statistics, contribution to EVA-TRISP in general and intellectual input in details of the design or the analyses of the research project; and lastly intellectual input to the writing and improving of the manuscript. These criteria are suggestions, and the researches taking the lead in each project take the final responsibility for the fair distribution of authorships.

Each member site possesses its own data, and each member site whose data are used is entitled to coauthorship(s). Whenever feasible, an abstract approved by all coauthors is submitted to the forthcoming ESO Conference. All EVA-TRISP member centres and investigators are listed at the end of the manuscript as a supplement given that the publishing journal's own format allows this approach.

### Data ownership, access, use and publications

Each centre is self-financing in data collection and is the indisputable owner of their own data. If and when a mutual single databank is constructed, each individual researcher with the necessary formal training and permissions will be able to insert data directly to the central database on the internet and have access to the researcher's own centre's data without limitations. Individual scientists working on a properly agreed on single project and doing data analyses will be granted proper access to all data. In case that an individual centre refrains from participating in a particular analysis, their data will not be included in that analysis. Any publication that is produced from the registry data will include authors from each contributing centre in accordance to number of patients delivered as well as active involvement in analyses and writing work. Number of authors and their placement in the author list may vary according to the amount of contributions. This will be handled openly and in a delicate way aiming at mutual consent.

### Ethics, informed consent and privacy

Each centre has received necessary official approval from their respective local authorities and/or ethical committees according to their national and local rules (online supplemental appendix 3). These permits include transfer of data between EVA-TRISP centres. Necessity of individual informed consent is dependent on national rules and will be collected if necessary. Data are shared with respect to the EU law 2016/679 about General Data Protection Regulation. In the long run, the aim is to have a permanent database residing at a member site. Establishment and maintenance of the permanent database at

one centre will be initiated only after a separate ethics approval.

## DISCUSSION

Endovascular treatment, now fulfilling the criteria for the highest level of evidence, has changed acute stroke care substantially. Probably approximately 10% of all ischaemic stroke patients are eligible for EVT, but the percentages may grow as more and more patients are brought to the attention of emergency systems, and the treatment indications will likely expand over time.[28] The rapid developments in acute stroke care put considerable demands on healthcare systems and necessitate quick rearrangements for coupling these needs. The seven published RCTs and following meta-analyses answered most central questions. Nevertheless, there are numerous unanswered questions remaining in terms of EVT in AIS. Some of these questions will be solved and satisfied via ongoing and forthcoming RCTs. Yet, many other issues will never or unlikely be tested in RCTs, and stroke physicians need firm data on these topics to base their clinical decisions on. Moreover, there are certain patient groups where RCTs are ethically difficult to organise; an example of this group is patients with basilar artery occlusion (BAO). Although a clearly important clinical condition that untreated has a poor outcome, BAO patients were not included in the large EVT trials. An extrapolation from the anterior circulation EVT trial results for BAO has currently a strong support in clinical practice, and thus EVT is currently offered to BAO patients despite limited direct evidence of treatment effectiveness. A small multicentre RCT that included 131 patients—the Chinese Basilar artery occlusion Endovascular intervention versus Standard medical Treatment trial[29]–was prematurely terminated due to slow recruitment and a high cross-over rate that severely hampered the interpretability of the intent-to-treat analysis. Another RCT with 300 patients included is closed, but data have not yet been presented.[30] Still, both the per-protocol and the as-treated analyses favoured EVT compared with best medical treatment. Some small-sized registry data showed high recanalisation rates and similar haemorrhagic complication rates as in anterior circulation patients treated with EVT but more often futile recanalisation. Large-scale registry studies may further improve our knowledge in this patient group and may help identify those who will likely benefit or not benefit from EVT in a real-life setting.[6 31–33] In the absence of RCT-based data, comprehensive observational data may be useful for individual treatment decisions in clinical practice and in evaluating processes of stroke triage and care for IVT or EVT. As a prerequisite, such data have to be based on well-maintained registries containing large numbers of detailed, clearly defined and well-characterised variables. The EVA-TRISP registry meets these prerequisites. Ideally, the results from such observational studies are verified or falsified by RCTs. However, with few exceptions (eg, age limit), this is unlikely to happen. Thus, registry-based data

will reflect the highest level of evidence in several aspects, available currently and in the foreseeable future.

Furthermore, we need to continuously follow-up whether the safety and benefit aspects of EVT shown in RCTs could be correctly translated to routine clinical practice. Indeed, the safety and benefit may be better, similar or even worse in daily practice. Registry data can easily be compared with RCT data especially when basic settings are similar. Additionally, it becomes more and more feasible to compare centres, patient subgroups and devices. Benchmarking, previously performed by site visits, is a popular approach for understanding differences and making improvements and can now be done easily using electronic data.[34]

Systematically ascertained, comprehensive and high-quality observational data are useful to both: (1) challenge or (2) confirm the clinical usefulness of commonly used but often arbitrary eligibility criteria. An early example has been the challenge of the usefulness of the upper age limit of 80 years for IVT based on comprehensive, observational studies. Eventually, the third International Stroke Trial proved that indeed patients aged 80 years and older benefit from IVT too.[35]

Using the TRISP registry, we previously examined the safety of IVT in a number of patient subgroups where RCT-based data did not exist. Previous publications of the TRISP registry: (1) provided insight into safety and efficacy of IVT in subgroups of patients who were excluded in RCTs (eg, patients dependent on the help of others prior to stroke), under-represented or not specifically addressed (eg, dissection as cause, impaired renal function, low platelet count, body mass index, prior use of statins, serotonin uptake inhibitors, prior use of novel oral anticoagulants and patients with seizure at onset)[21 36–45]; (2) facilitated the evaluation process of acute stroke care such as the meaning of the 'off-hour-thrombolysis', IVT during 'working hours' or the variable 'time' in clinical practice[46–48]; and (3) served to derive, validate and compare risk scores for sICH or functional 3 months outcome.[49–51] These registry-based novel data contributed to the numbers of patients treated safely and successfully with IVT globally. Ongoing and planned research projects within the EVA-TRISP registry collaboration that may fill important knowledge gaps are investigations on: (I) stroke due to cervical artery dissection, (2) stroke with low baseline NIHSS, (3) stroke specifically in the anterior cerebral artery (ACA) territory, (4) stroke patients with preexisting dependency, (5) significance of cerebral collaterals, (6) significance of tandem occlusions and (7) stroke patients with active cancer.

Disease-based or intervention-based patient registries with consecutive patients recruited in a population-based or hospital-based approach are useful in many ways: they help describe the natural history, determine clinical effectiveness and cost-effectiveness of healthcare products or services, measure or monitor safety and harm, measure quality of care, improve quality of care and help with benchmarking purposes such as how clinical practices vary, what the best predictors of treatment practices are and comparing different practices providing a basis for further improvements. In such settings, stakeholders are several: the primary stakeholder with the EVA-TRISP registry is the academic consortium establishing and running the registry. Potential stakeholders with such a large-scale registry may include public health and regulatory authorities, product manufacturers, healthcare service providers, payer and commissioning authorities, patients and their advocacy groups, treating physician groups, academic institutions and professional societies. The EVA-TRISP registry aims at including all patients who underwent EVT as a treatment for AIS, including patients with misdiagnosis, unsuccessful attempts (EVT is defined as a puncture to the artery with the aim of recanalisation) or other unforeseeable scenarios. In most countries, registry-based studies are approved by ethics committees with waving informed consent from individual patients as demanding informed consent would leave most severe patients out of the registry and cause a severe bias on representability of any finding.

Imaging has become more and more critical in the stroke field. In addition to stroke diagnostics and excluding competing aetiologies (eg, stroke mimics), there are a number of imaging findings related to increased risk following acute treatments (eg, leukoaraiosis[52] and microbleeds), or findings guiding treatment choices (eg, major artery occlusion reachable with a catheter), as well as findings helpful in prognostics such as ASPECTS[53] or blood Suger, Early ischemic changes, hyperDens artery sign, Age, NIHSS (SEDAN)[49] scores. Acute stroke patients are increasingly imaged with a package of standard CT, CT angiography and CT perfusion or in a similar fashion with an MRI-based package. These imaging modalities are then analysed quickly for determining diagnosis, prognosis and treatment approach. Functional imaging modalities are increasingly taken into account for patient selection to IVT and EVT instead of strictly deciding according to time. The former TRISP registry included few items on imaging studies. Imaging requirements were according to routine thrombolysis: only a non-contrast CT imaging prior to thrombolysis was mandatory. Imaging was not the main focus as image analysis is usually labour intensive and not always available. Technological improvements may substitute some of the expert workforce in image analysis already today and in the near future. However, with the developments in imaging technologies and increased requirements in patient care, currently most patients are imaged with CT angiography and CT perfusion in addition to basic non-contract CT imaging. Less frequently, patients are imaged with a similar versatile package of various MRI sequences. Developments in the imaging technology and logistics, decrease in radiation dose used with CT imaging, as well as automatic image analysis software development contributed to the progress. Installing imaging scanners into or adjacent to emergency departments or taking acute stroke suspect patients directly to the imaging facility by-passing the emergency room has also improved the availability of more detailed imaging. Patient selection for the best individual treatment

is becoming increasingly dependent on neuroimaging with the goal of providing rapid patient-specific metrics such as tissue viability, vessel patency status, thrombus characteristics and cerebral perfusion, etc. Imaging findings have also been used for patient selection in some highly successful IVT and EVT RCTs.[4 5 54–56] Detailed imaging information is even more crucial when EVT is considered. Therefore, the establishing of an imaging repository parallel with the EVA-TRISP registry received a widespread support from members. To date, the choice of imaging modalities, parameters and thresholds varies widely across medical centres. No standardised imaging protocols currently exist, other than joint statements from professional societies.[57] A large, multicentre neuroimaging registry with state-of-the-art re-evaluation of images combined with detailed clinical data of IVT/EVT-treated stroke patients would be helpful for validating between modalities, defining thresholds, enhancing automated assessments and creating standards in neuroimaging for AIS. For the imaging part of the database, we will collect baseline, interventional and follow-up images (up to 3 months after stroke onset) from all stroke patients included in TRISP since 2012. All images will be centrally analysed using a predefined, standardised form.

### Strengths and limitations of the EVA-TRISP registry

Strengths of EVA-TRISP registry include: (1) the high completeness level of data with few missing data, (2) large sample sizes that reduce the risk of bias and allows adjustments for confounders, (3) the systematic and standardised data ascertainment that increases the homogeneity of the study population, (4) the intrinsic motivation of the study personnel that leads to a high rate of completeness of ascertained data sets, contributing to a high-quality registry and (5) the dynamic nature of the EVA-TRISP database due to the commitment of the centres to adapt the local database and add variables retrospectively and prospectively. In addition, (6) a large number of variables is gathered including those with unknown prognostic importance. This allows addressing novel yet unidentified research questions. Moreover, (7) pooling of individual patient data increases generalisability compared with single centre studies, and (8) the fact that variables and outcomes have been collected irrespective of the present research question, reduces the risk of a bias. (9) As most EVA-TRISP centres are regional reference centres for acute patient care, particularly for EVT, the EVA-TRISP registry will resemble a population-based registry. Limitations are inherent to the design of EVA-TRISP: (1) Data are derived from registries that are neither monitored nor randomised. Usually, there will be no control group without EVT that disallows the assessment of effectiveness of EVT in study populations. (2) As true for all observational studies, analyses based on registers have a higher risk of bias than RCTs. Thus, we urge to a cautious interpretation of findings and observations. (3) All EVA-TRISP centres are experienced in stroke treatment. This expertise implies—as a downside—a limited generalisability of findings to all stroke providers with less expertise and less advanced settings. (4) The majority of our included patients are Caucasians and from high-income countries. Thus, we cannot compare ethnical differences, nor can we compare health systems with various funding levels. (5) Currently, there is no 'core lab' to validate haemorrhagic complications and 3-month mRS ratings. As valid for other major registries like SITS and GWTG, local interpretation of outcome data may differ between sites. Since EVA-TRISP centres are mostly high-volume centres with long-standing experience in maintaining IVT databases, this bias is likely to be smaller than in most of the other registries.

### SUMMARY

The EVA-TRISP collaboration is an open platform dedicated to conduct joint research projects in AIS patients treated with IVT and/or EVT. The aims of the EVA-TRISP collaboration are to increase knowledge on the safety and efficacy of IVT and EVT, study outcomes after IVT and EVT, to evaluate processes of acute stroke care as well as to document and improve acute stroke care quality. Our previous achievements prove that this collaboration has the potential to provide versatile observational information on treatment of AIS patients during daily clinical practice. Prospective and standardised documentation of individual patient data according to consensus definitions is a major requirement to maintain the quality of the EVA-TRISP registry. Publishing this methodology paper improves the transparency of the registry and collected data. EVA-TRISP welcomes participation and project proposals of further centres fulfilling the requirements stated previously.

**Author affiliations**
[1]Department of Clinical Neurosciences Sahlgrenska Academy at University of Gothenburg, Sahlgrenska Academy, Goteborg, Sweden
[2]Department of Neurology, Sahlgrenska University Hospital, Gothenburg, Sweden
[3]Department of Neurology, University of Helsinki and Helsinki University Hospital, Helsinki, Finland
[4]Department of Neurology and Stroke Center, University of Basel and University Hospital Basel, Basel, Switzerland
[5]Department of Neurology, Amsterdam University Medical Centres, Amsterdam, The Netherlands
[6]Department of Neurology and Center for Stroke Research, Charité Universitätsmedizin Berlin, Berlin, Germany
[7]Department of Neurology and Center for Stroke Research, Charite Universitatsmedizin Berlin Klinik fur Neurologie mit Experimenteller Neurologie, Berlin, Germany
[8]Department of Neurology, University of Lille, Lille, France
[9]Department of Neurology, Centre Hospitalier Universitaire Vaudois and University of Lausanne, Lausanne, Switzerland
[10]Department of Neurology, University Hospital Heidelberg, Heidelberg, Germany
[11]Department of Neurology, Inselspital, Bern University Hospital, University of Bern, Bern, Switzerland
[12]Department of Clinical and Experimental Sciences, Neurology Clinic, University of Brescia, Brescia, Italy
[13]Department of Neurology and Stroke Center, Maggiore Hospital, IRCCS Istituto Delle Scienze Neurologiche di Bologna, Bologna, Italy
[14]Neurology Clinic, Clinical Centre of Serbia, Belgrade, Clinical Center of Serbia, Beograd, Serbia
[15]Faculty of Medicine, University of Belgrade, Belgrade, Serbia
[16]Department of Neurology, University of Zurich, Zurich, Switzerland
[17]Department of Neurology, Ludwig-Maximilians-Universitat Munchen, Munchen, Germany
[18]Neurology and German Center for Vertigo and Balance Disorders, Ludwig-Maximilians-Universitat Munchen, Munchen, Germany

[19]Department of Neurology, Kantonsspital St Gallen, Sankt Gallen, Switzerland
[20]Department of Radiology, Institute of clinical sciences, Sahlgrenska Academy, University of Gothenburg, Goteborg, Sweden
[21]Diagnostic and Interventional Neuroradiology, Sahlgrenska University Hospital, Gothenburg, Sweden
[22]Department of Radiology, Helsinki University Central Hospital, Helsinki, Finland
[23]Department of Neurology, Hebrew University Hadassah Medical School, Yerushalayim, Israel
[24]Department of Neurosurgery, Hebrew University Hadassah Medical School, Yerushalayim, Israel
[25]Department of Radiology, Hebrew University Hadassah Medical School, Yerushalayim, Israel
[26]Department of Neurology, Georg-August-Universitat Gottingen Universitatsmedizin, Gottingen, Germany
[27]Department of Neuroradiology, Georg-August-Universitat Gottingen Universitatsmedizin, Gottingen, Germany
[28]Neuroradiology and Stroke Center Basel, University of Basel and University Hospital Basel, Basel, Switzerland
[29]Department of Neurology, University Hospitals Bremen-Mitte and Bremen-Ost, Bremen, Germany
[30]Clinic for Diagnostic and Interventional Neuroradiology, University Hospitals Bremen-Mitte and Bremen-Ost, University of Bremen, Bremen, Germany
[31]Department of Radiology, Areteion University Hospital, National and Kapodistrian University of Athens, Athens, Greece
[32]Institute of Diagnostic and Interventional Neuroradiology, Inselspital, Bern University Hospital, University of Bern, Bern, Switzerland
[33]USD Stroke Unit and Vascular Neurology, ASST Spedali Civili di Brescia, Brescia, Italy
[34]Department of Neuroradiology, Amsterdam University Medical Centers, Amsterdam, The Netherlands
[35]Institute of Neuroradiology, Charité-Universitätsmedizin, Berlin, Germany
[36]Clinic of Neurosurgery, Clinical Center of Serbia, Beograd, Serbia
[37]Faculty of Medicine, University of Belgrade, Beograd, Serbia
[38]Clinic for Vascular and Endovascular Surgery, Clinical Centre of Serbia, Belgrade, Serbia
[39]Department of Radiology and Nuclear Medicine, Kantonsspital St Gallen, Sankt Gallen, Switzerland
[40]Department of Neuroradiology, University Hospital Zurich, Zurich, Switzerland
[41]Department of Neuroradiology, UniversitatsKlinikum Heidelberg, Heidelberg, Germany
[42]Department of Internal Medicine, Faculty of Medicine, School of Health Sciences, University of Thessaly, Volos, Thessaly, Greece
[43]Department of Radiology, University Hospital of Larissa, School of Medicine, University of Thessaly, Volos, Thessaly, Greece
[44]Department for Clinical Neuroscience and Rehabilitation, Sahlgrenska Academy, Goteborg, Sweden
[45]Department of Neurology and Stroke Center, University of Basel, Basel, Switzerland
[46]Deptartment of Radiology, Areteion University Hospital, National and Kapodistrian University of Athens, Athen, Greece
[47]Department of Clinical Neurosciences, Sahlgrenska University Hospital, Goteborg, Sweden
[48]Department of Neurology, University of Helsinki, Helsinki, Finland

**Acknowledgements** We thank Anu Eräkanto and Judith Klecki for technical support.

**Collaborators** EVA-TRISP Investigators.

**Contributors** Data collection: all authors. Manuscript drafting: AN, SC, HG, SMZ, HE, CK, CHN, PJN, KJ, STE, DS and TT. Study supervision: TT, DS, STE, KJ, PJN and CHN. Statistical analysis and interpretation: does not apply. Review of the manuscript for intellectual contribution: all authors. All authors agreed on submitting this last version of the manuscript to the journal.

**Funding** The authors have not declared a specific grant for this research from any funding agency in the public, commercial or not-for-profit sectors.

**Map disclaimer** The inclusion of any map (including the depiction of any boundaries therein), or of any geographic or locational reference, does not imply the expression of any opinion whatsoever on the part of BMJ concerning the legal status of any country, territory, jurisdiction or area or of its authorities. Any such expression remains solely that of the relevant source and is not endorsed by BMJ. Maps are provided without any warranty of any kind, either express or implied.

**Competing interests** None declared.

**Patient consent for publication** Not required.

**Provenance and peer review** Not commissioned; externally peer reviewed.

**Data availability statement** No data are available. All data relevant to the study are included in the article or uploaded as supplementary information. Not applicable.

**Author note** KJ, CHN, PJN, STE, DS and TT are joint last authors.

**ORCID iDs**
Annika Nordanstig http://orcid.org/0000-0002-3625-2168
David J Seiffge http://orcid.org/0000-0003-3890-3849
Katharina Feil http://orcid.org/0000-0002-4566-712X
Christian H Nolte http://orcid.org/0000-0001-5577-1775

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
