## [Reviewer comments · BMJ Open]

ARTICLE DETAILS

TITLE (PROVISIONAL)	Cohort profile: EndoVascular treatment and Thrombolysis for Ischemic Stroke Patients (EVA-TRISP) Registry: Basis and methodology of a pan-European prospective ischemic stroke revascularization treatment registry
AUTHORS	Nordanstig, Annika; Curtze, Sami; Gensicke, H; Zinkstok, Sanne; Erdur, Hebung; Karlsson, Camilla; Karlsson, Jan-Erik; Martinez-Majander, Nicolas; Sibolt, G; Lyrer, Philippe; Traenka, Christopher; Baharoglu, Merih; Scheitz, Jan; Bricout, Nicolas; Hénon, Hilde; Leys, Didier; Eskandari, Ashraf; Michel, Patrik; Hametner, Christian; Ringleb, Peter; Arnold, Marcel; Fischer, Urs; Sarikaya, Hakan; Seiffge, David; Pezzini, Alessandro; Zini, Andrea; Padjen, Visnja; Jovanovic, Dejana R.; Luft, Andreas; Wegener, Susanne; Kellert, Lars; Feil, Katharina; Kägi, Georg; Rentzos, Alexandros; Lappalainen, Kimmo; Leker, RR; Cohen, Jose; Gomori, John; Brehm, Alex; Liman, Jan; Psychogios, Marios; Kastrop, Andreas; Papanagiotou, Panagiotis; Gralla, Jan; Magoni, Mauro; Majoie, Charles; Bohner, Georg; Vukasinovic, Ivan; Cvetic, Vladimir; Weber, Johannes; Kulcsar, Zolt; Bendszus, Martin; Möhlenbruch, Markus; Ntaios, George; Kapsalaki, Eftychia; Jood, Katarina; Nolte, Christian; Nederkoorn, Paul J; Engelter, Stefan; Strbian, Daniel; Tatlisumak, T

VERSION 1 – REVIEW

REVIEWER	Tian, Huiqiao University of New South Wales - Kensington Campus
REVIEW RETURNED	24-Aug-2020

GENERAL COMMENTS	The EVA-TRISP registry will allow multiple collaborations in stroke research. Also, the authors state that new proposals will be reviewed and discussed internally before study initiation/data collection. This will hopefully provide a good quality of future research from the registry. The manuscript is well written.
--

REVIEWER	Rai, Ansaar WEST VIRGINIA UNIVERSITY, NEURORADIOLOGY
REVIEW RETURNED	25-Sep-2020

GENERAL COMMENTS	The authors present an elaborate outline of a potential or upcoming multi-center, multi-national registry for intravenous thrombolysis and endovascular therapy of AIS. The authors describe the aims and methods of the registry to collect data from multiple centers that can then be used to answer stroke related questions. The concept of a multi-center registry to collect prospective data is a good one but not novel and similar registries exist. The manuscript describes what data will / could be collected
---

	and how it will / could be used. These are all valid and relevant goals. At this point though just the description of how the process will work is of limited interest. It would be good to have some concrete hypothesis driven projects that have been defined and even better to see the results. Just the description of how this data will be collected and what could be done without more refined ideas is I am afraid lacking in detail.
--	--

REVIEWER	Ramkumar, Niveditta Dartmouth-Hitchcock Medical Center
REVIEW RETURNED	21-Feb-2021

GENERAL COMMENTS	The authors describe the protocol for developing a multicenter registry to study patients undergoing endovascular treatment for acute ischemic stroke. The registry will build off of the already established TRISP registry. The successful implementation of this registry can allow for developing evidence on research questions that cannot be answered with RCTs. A few comments: Page 20, line 22: can the authors clarify what they mean by “completed by reconstructing”? Page 21, lines 4- page 33 line 38: authors should include some more discussion of quality control of the data collected at the individual sites- who will perform it? Is there a comprehensive list of expert-informed checks? Some are listed in the text here, but I think it would be helpful to see this in another appendix to attest to data quality. Also, what are the next steps if there are errors discovered in the collected data? Page 22, line 49-51: Authors state the informed consent will be obtained if necessary- can you elaborate a little more on what that would look like for this patient population with acute ischemic stroke? Page 23, line 12: I think this section about the aims of the EVA-TRISP registry would fit better earlier in the protocol, perhaps immediately after the introduction. I think the different sections between the introduction and discussion can be rearranged to improve flow. I’d suggest starting with the aims, detailing data collection and quality controls, and then dive into authorship/publication (or how the data will be used). Consider using a table to highlight the strengths and limitations of EVA-TRISP across key areas compared to other preexisting EVT registries (as alluded to on page 14, line 49). There are minor grammatical and spelling errors scattered across the manuscript. Would recommend proof-reading.
--

VERSION 1 – AUTHOR RESPONSE

Reviewer: 1
Dr. Huiqiao Tian, University of New South Wales - Kensington Campus
Comments to the Author:

The EVA-TRISP registry will allow multiple collaborations in stroke research. Also, the authors state that new proposals will be reviewed and discussed internally before study initiation/data collection. This will hopefully provide a good quality of future research from the registry. The manuscript is well written.

Response: We appreciate this comment from the reviewer.

Reviewer: 2

Dr. Ansaar Rai, WEST VIRGINIA UNIVERSITY

Comments to the Author:

The authors present an elaborate outline of a potential or upcoming multi-center, multi-national registry for intravenous thrombolysis and endovascular therapy of AIS. The authors describe the aims and methods of the registry to collect data from multiple centers that can then be used to answer stroke related questions. The concept of a multi-center registry to collect prospective data is a good one but not novel and similar registries exist. The manuscript describes what data will / could be collected and how it will / could be used. These are all valid and relevant goals. At this point though just the description of how the process will work is of limited interest. It would be good to have some concrete hypothesis driven projects that have been defined and even better to see the results. Just the description of how this data will be collected and what could be done without more refined ideas is I am afraid lacking in detail.

Response: We thank the reviewer for this relevant comment and thank the reviewer for the review which enabled us to improve our manuscript further. Although the main purpose of this EVA-TRISP registry position paper is to comprehensively describe the registry purpose and infrastructure for the data collection process, we agree with the reviewer that some examples of projected and ongoing research projects within the EVA-TRISP collaboration may add value by increasing the understanding by readers what will be achievable in this particular registry collaboration. We therefore added the following text to the discussion part of the paper (page 23, lines 624-628):

“Ongoing and planned research projects within the EVA-TRISP registry collaboration that may fill important knowledge gaps are investigations on (I) stroke due to cervical artery dissection, (II) stroke with low baseline NIHSS, (III) stroke specifically in the ACA-territory, (IV) stroke patients with preexisting dependency, (V) significance of cerebral collaterals, (VI) significance of tandem occlusions and (VII) stroke patients with active cancer.”

Reviewer: 3

Dr. Niveditta Ramkumar, Dartmouth-Hitchcock Medical Center

Comments to the Author:

The authors describe the protocol for developing a multicenter registry to study patients undergoing endovascular treatment for acute ischemic stroke. The registry will build off of the already established TRISP registry. The successful implementation of this registry can allow for developing evidence on research questions that cannot be answered with RCTs.

Response: We appreciate this comment from the reviewer.

A few comments:

Page 20, line 22: can the authors clarify what they mean by “completed by reconstructing”?

Response: one of the unique features of the EVA-TRISP registry collaboration is that the collection of agreed on registry variables is undertaken by dedicated stroke treatment health care professionals that are highly involved both in the clinical care of stroke patients during their entire treatment process and in the EVA-TRISP collaboration. Hence, the EVA-TRISP collaborators at the different participating stroke centers will, as far as possible, safeguard that all registry variables are registered and introduced in the local registry in a prospective and timely manner. It may however be that in some instances the stroke physician will find that, although a comprehensive neurological exam have been undertaken in the emergency ward, the actual NIHSS score may not have been documented in the medical record and during such circumstances the EVA-TRIPS stroke physician will then reconstruct the score based on the documented neurological exam. This is what we meant by this statement. However, we realize that our description regarding this in the text in the original submission might have been somewhat misleading to the reader why we in the revised manuscript have changed wording for this particular section, as follows (page 16 lines 461-468):

“All paper-based or electronic patient files including laboratory values and imaging data will be utilized to capture the EVA-TRISP registry data points. Based on these different sources alongside repeated clinical evaluations undertaken by the dedicated EVA-TRISP collaborator, registry data points that initially remain missing during the early stroke treatment process will in the majority of cases be possible to reconstruct and thereafter reported to the local registry (e.g. NIHSS scores). In most cases, the local EVA-TRISP investigators form the local stroke team and will be actively seeing the patients already in the emergency room and/or at their own stroke units and can therefore guarantee completeness of data in most cases.”

Page 21, lines 4- page 33 line 38: authors should include some more discussion of quality control of the data collected at the individual sites- who will perform it? Is there a comprehensive list of expert-informed checks? Some are listed in the text here, but I think it would be helpful to see this in another appendix to attest to data quality. Also, what are the next steps if there are errors discovered in the collected data?

Response: We agree with the reviewer that appropriate steps for quality control is very important in any clinical registry and already in the original submission we therefore provided a brief description of the EVA-TRISP measures for quality control. To meet the requirements of the reviewer, in the revised paper we have elaborated further on our process for quality controls, as cited directly below (page 16-17 lines 425-451). However we suggest that this important section is kept within the main body of the manuscript rather than introducing it as another appendix, and hope that the below clarified section will be acceptable to the reviewer. In the below revised section we have specifically tried to answer the reviewer’s points about who do the quality checks.

“Quality Control:

Quality control is another crucial step in multicenter large-sized registries since missing data are a frequent problem impairing the reliability and generalizability of registry-borne data. The EVA-TRISP registry is different in this sense because data are not yet directly collected to a central registry, but each center collects their own data to their own institutional registry according to a standard harmonized database item list and SOP. Thus, clear variable definitions are easily available for the user when data are entered. It is the responsibility of the local EVA-TRISP collaborator (usually a senior stroke physician) to check and account for the validity and completeness of all data points introduced to the local registry. Therefore, missing data are expected to be very low. Furthermore, all centers have agreed to include all consecutive patients attempted with an EVT and all centers will undertake frequent spot checks against internal hospital administrative systems covering EVT procedures, as to not leave any patient out of the registry. Therefore our registry data will likely include all EVTs performed within a region and population, practically equaling to a population-based study, although being hospital-based, because EVT is usually available only at stroke centers serving

a predefined region and the inhabitant population. Most of the required data come from routine procedures which are standardly collected and recorded in stroke patient care pathways as part of the clinical routine and therefore these data points are almost always retrievable. In the subsequent quality control process, the data files from the individual participating centers are merged into a single file for further maintenance analyses. Pseudonymized individual center data are sent to the center leading the specific project using encrypted transfer protocols. The subsequent data management of the merged database will implement checks for missing data along with checks for range, consistency and illogical data (e.g. NIHSS score cannot be minus or over 42 points and can only be full points and not decimals; patient age at stroke onset can be only in digits, and is expected to be from 16 and very rarely over 100). Also other procedures regarding quality check will be implemented at milestone points, such as comparing the performance of data reporting among centers.”

Page 22, line 49-51: Authors state the informed consent will be obtained if necessary- can you elaborate a little more on what that would look like for this patient population with acute ischemic stroke?

As national laws and other rules regulating clinical research differ in each country and sometimes even within the same country in different regions, it is expected that some centers will be required to receive informed consent while at some other centers ethics committees will approve the registry waiving the need for an informed consent. Also, the rules change over time and sometimes unpredictably. Therefore, we keep door open for different scenarios. However, we are aware of the hindrances of having informed consent in acute stroke patients. Often times, but not always, these difficulties are taken into consideration by ethic committees. After hand informed consent, opt-out for registry studies and next-to-kin informed consent approaches are also commonly utilized in different countries in Europe.

If informed consent is required beforehand, it will inevitably drop most of the severely ill patients leading to a skewness of data towards mild strokes. Additionally, data collection would not be consecutive, a weakness in data-quality. We will require ethics committee approval from each participating site, and the enclosure of ethics committee approval will definitely differ between centers.

Page 23, line 12: I think this section about the aims of the EVA-TRISP registry would fit better earlier in the protocol, perhaps immediately after the introduction. I think the different sections between the introduction and discussion can be rearranged to improve flow. I'd suggest starting with the aims, detailing data collection and quality controls, and then dive into authorship/publication (or how the data will be used). Consider using a table to highlight the strengths and limitations of EVA-TRISP across key areas compared to other preexisting EVT registries (as alluded to on page 14, line 49).

Response: Thanks for this comment. We agree with the reviewer and therefore we have moved the section “aims of the EVA-TRISP registry” to a position immediately after the introduction. As also suggested, we rearranged the relevant sections between introduction and discussion in order to further improve flow and readability. We however did not implement a table comparing strengths and imitations for the EVA-TRISP registry as compared to other remotely comparable stroke registries. The main reason for this is that it is often far from clear in the literature how these potentially comparable EVT registries are organized, what quality control measures that are run, and how the entire logistic process for data collection are organized etc. We therefore feel that we cannot with enough precision describe the strengths and limitations of other available EVT registries as compared to ours. We hope that the reviewer can understand this position. In fact, a clear and transparent description of the EVA-TRISP registry collaboration was the main aim of our current paper and it is not unusual that such information is largely lacking for comparable stroke/EVT registries.

There are minor grammatical and spelling errors scattered across the manuscript. Would recommend proof-reading.

Response: we have again proof-read our revised manuscript and corrected identified spelling errors.